# Two New Phoretic Species of Heterostigmatic Mites (Acari: Prostigmata: Neopygmephoridae and Scutacaridae) on Australian Hydrophilid Beetles (Coleoptera: Hydrophilidae) [note 1]

**DOI:** 10.3390/insects13050483

**Published:** 2022-05-22

**Authors:** Hamid Khadem-Safdarkhani, Hamidreza Hajiqanbar, Markus Riegler, Owen Seeman, Alihan Katlav

**Affiliations:** 1Department of Entomology, Faculty of Agriculture, Tarbiat Modares University, Tehran P.O. Box 14115-111, Iran; hamid.khadem@modares.ac.ir; 2Hawkesbury Institute for the Environment, Western Sydney University, Locked Bag 1797, Penrith, NSW 2751, Australia; m.riegler@westernsydney.edu.au; 3Queensland Museum, South Brisbane, QLD 4101, Australia; owen.seeman@qm.qld.gov.au

**Keywords:** systematics, Australia, symbiosis, mites, aquatic insects

## Abstract

**Simple Summary:**

Heterostigmatic mites are globally very diverse, and many are generally phoretic or parasitic on insects. However, the Australian fauna of phoretic heterostigmatic mites is almost unexplored. Here, we describe two new species of *Allopygmephorus* and *Archidispus* (Acari: Prostigmata: Heterostigmata) found phoretic on semiaquatic beetles. Our findings report both genera for the first time from Australia.

**Abstract:**

Many heterostigmatic mites (Acari: Prostigmata: Heterostigmata) display a wide range of symbiotic interactions, from phoresy to parasitism, with a variety of insects. Australia is expected to harbour a rich diversity of heterostigmatic mites; however, its phoretic fauna and its host associations remain mainly unexplored. We conducted a short exploration of Australian insect-associated phoretic mites in summer 2020 and found two new phoretic heterostigmatic species on a semiaquatic hydrophilid beetle species, *Coelostoma fabricii* (Montrouzier, 1860) (Coleoptera: Hydrophilidae). Here, we describe these two new species, *Allopygmephorus coelostomus* sp. nov. (Neopygmephoridae) and *Archidispus hydrophilus* sp. nov. (Scutacaridae), which both belong to the superfamily Pygmephoroidea. Both species are distinct from their congeners, with a plesiomorphic character, bearing a median genital sclerite (mgs). Our study reports both genera for the first time from Australia.

## 1. Introduction

Heterostigmatic mites (Acariformes: Prostigmata) are morphologically diverse, and numerous species are associated with a variety of insects [1,2]. Their associations with insects vary from facultative or obligate phoresy to parasitism [1]. Many species are free-living, and sometimes hitchhike on insects to new environments, or reside in nests of social insects [2]. Some of them are highly host-specific and they sometimes exhibit special attachment site preferences on the hosts’ body, such as beneath the abdomen or thorax, between leg coxae, on cervix or wings, or under elytral spaces [3,4,5]. Females in phoretic species have generally evolved several morphological adaptations, such as a robust pair of legs I with consolidated tibiotarsi I associated with enlarged claws, which help them to firmly attach to their hosts during phoretic dispersal [1,6].

The superfamily Pygmephoroidea is one of the most diverse groups of Heterostigmata, with more than 1200 species within four families, Microdispidae Cross, 1965, Neopygmephoridae Cross, 1965, Pygmephoridae Cross, 1965, and Scutacaridae Oudemans, 1916 [7,8]. Mites of this superfamily are generally fungivorous and many are generally found to be phoretic on beetles, bees, ants and termites [2,9,10,11,12,13] and, less frequently, on other arthropods or small mammals [13,14]. Exceptions are a few species of the family Microdispidae, which might be parasitoids of the insects on which they are phoretic [1]. Neopygmephorid mites encompass 26 genera with about 300 species [7,15,16], with a high number of mite species inhabiting nests of a wide range of ant species [17,18]. Many others are phoretic on saproxylic or coprophagous beetles [19,20,21]. Scutacarid mites include more than 24 genera with more than 800 species [7] that are very small (from 200 to 400 µm) in length and are typically recognized by their hemispherical shape. These mites are widely distributed in soil, decaying material, moss and humus [22]. Many of them display sexual dimorphism and sometimes phoretic morphs, in which only phoretic females have enlarged claws on legs I, allowing them to cling firmly to their host (mainly beetles, bees and ants) for phoretic dispersal [23,24].

Several heterostigmatic species favor moist habitats, and these are generally found on insects inhabiting semiaquatic habitats such as riverbanks, flood swamps and shorelines [25]. For example, the scutacarid genera *Archidispus* Karafiat, 1957, *Imparipes* Berlese, 1903, *Pygmodispus* Paoli, 1911 and *Scutacarus* Gros, 1845 can sometimes be found under the elytra or on other body surfaces of semiaquatic beetles. The genus *Archidispus* is predominantly phoretic on carabid beetles [9,26,27,28,29,30]; however, a handful of species have been recorded from Heteroceridae MacLeay, 1825, Hydrophilidae Latreille, 1802, and Staphylinidae Latreille, 1802, mainly dwelling in moist habitats [24]. Likewise, among neopygmephorids, the genus *Allopygmephorus* is well known for its members being hydrophiles and having evolved relationships with insects occupying moist habitats.

Although it has been predicted that Australia is home to a large diversity of heterostigmatic mites, with recent studies discovering a rich diversity of parasitic fauna [4,31,32], the phoretic fauna in Australia remain almost unexplored. The only reported Australian Pygmephoroidea were found from soil samples, and their phoretic associations with host insects are mostly unknown. For example, apart from *Scutacarus hydrophilus* Mahunka, 1967, which was found to be phoretic on phorid and sciarid flies, Australian scutacarid species (30 species belonging to the genera *Heterodispus* Paoli, 1911, *Diversipes* Berlese, 1903, *Imparipes*, *Pygmodispus* and *Scutacarus*) are only known from soil samples [31]. The neopygmephorid fauna is even less explored (five species belonging to the genera *Bakerdania* Sasa, 1961, *Pseudopygmephorus* Cross, 1965, *Troxodania* Khaustov and Trach, 2014) with only *Troxodania* troxi (Mahunka and Philips, 1977) having been found on an Australian trogid beetles [31,33,34]. Therefore, more studies are required to obtain a better understanding of the diversity of phoretic heterostigmatic mites in Australia and their host associations.

Following a short exploration of insect hosts for heterostigmatic mites in summer 2020, we found several specimens of a semiaquatic hydrophilid beetle species, *Coelostoma* (*Coelostoma*) *fabricii* (Montrouzier, 1860) (Coleoptera: Hydrophilidae) to be heavily co-infested with two new heterostigmatic mite species, one each of *Allopygmephorus* and *Archidispus*. Here, we describe and illustrate these two new species, *Allopygmephorus coelostomus* sp. nov. (Neopygmephoridae) and *Archidispus hydrophilus* sp. nov. (Scutacaridae) and report both genera for the first time for Australian mite fauna.

## 2. Materials and Methods

Mites were removed from their host beetles under a stereomicroscope using a fine brush. Subsequently, they were cleared in lactophenol, mounted in Hoyer’s medium and studied using a phase contrast microscope (model BX51, Olympus). All obtained specimens of both mite species were phoretic adult females attached to the host beetles’ sternite between the coxae. All specimens were collected by A. Katlav. The hydrophilid host beetles were identified with the help of Jason F. Mate (Department of Entomology, The Natural History Museum, London, UK) and Martin Fikacek (Department of Entomology, National Museum, Prague, Czech Republic). All measurements are given in micrometres for holotypes, as well as the measurements for five paratypes (in parentheses). Distances between setae were measured from the base of one seta to that of another; setae represented by the acetabulum only are designated as ‘vestigial setae’ and those as long as the acetabulum or shorter are designated as ‘microsetae’. Terminology and chaetotaxy follow Lindquist [35]. The family classification of Pygmephoroidea (Heterostigmatina) follows that of Khaustov [9].

The holotypes were deposited at the Queensland Museum, QLD, Australia. From each new species, two paratypes were deposited at the Acarological Collection, Department of Entomology, Faculty of Agriculture, Tarbiat Modares University, Tehran, Iran. The remaining mite and host beetle specimens were retained with the holotype.

## 3. Results

### 3.1. Family Neopygmephoridae Cross, 1965

Genus *Allopygmephorus* Cross, 1965.

Type species: *Pygmephorus matthesi* Krczal, 1959, by original designation.

**Diagnosis:** Adult female. Seta *v*2 absent, distance between pharyngeal pumps II and III short (less than half the length of pump III), setae *ps*2 absent, cupules ih absent, setae 1b thickened, lanceolate with a notch or slit from about middle of seta to distal, genital sclerites ags and pgs (and sometimes mgs) present.

*Allopygmephorus coelostomus* Katlav and Hajiqanbar **sp. nov.** (Figure 1, Figure 2, Figure 3 and Figure 4).

Type Material.

**Holotype:** female (QMS 120029), ex. under sternite between leg I and II coxae of *C. fabricii*; Loc. Vines Dr, Hawkesbury campus, Western Sydney University, Richmond, NSW, 33°36′45.6″ S 150°44′40.2″ E; coll. A. Katlav; 12 Feburary 2020.

**Paratypes:** Four females (QMS 120030-2; TMU SP202206-AK2), same data as holotype.

**Female.** Length of idiosoma 198 (165–196), width 157 (133–176).

*Gnathosoma* (Figure 1): length of gnathosoma 28 (26–28), width 23 (20–24). Gnathosomal capsule (Figure 1a) slightly longer than wide; dorsally with two pairs of subequal pointed and smooth cheliceral setae *cha* 16 (13–15) and *chb* 14 (11–14); postpalpal setae (*pp*) 9 (9–10) setiform, smooth and pointed; dorsal median apodeme not visible; subcapitulum (Figure 1b) with one pair of smooth and pointed subcapitular setae *su* 11 (10–11), distance between setae *cha–cha* 14 (13–14)*, chb–chb* 4 (4–5)*, su–su* 11 (10–11); palps dorsally bearing two subequal pointed and smooth setae *d*Fe 12 (11–12) and *d*Ge 10 (9–10), ventrally with an accessory setigenous structure (*ass*), palpi terminating in sclerotized falciform claws; pharyngeal pumps (Figure 1b and Figure 2a) including three striated pumps: pump 1 (php1) smallest bow-like, situated inside gnathosomal capsule; pump 2 (php2) strongly developed, sub-elliptical; pump 3 (php3) bow-like, horizontal rectangular, slightly bigger than pump 1. Connection between php2 and php3 not discernible.

*Idiosomal dorsum* (Figure 2a and Figure 4a): Anterior margin of tergite C covering posterior half of prodorsum; all dorsal plates ornamented with small dimples; stigmata subellipsoidal, associated with elongate atria with posterior tracheal branches; trichobothria clavate and smooth. All dorsal setae sharply pointed except seta *e,* which is distinctly stiff and blunt-ended. Setae *sc*2, *c*1, *c*2, *d* and *e* smooth, setae *f*, *h*1 and *h*2 with a couple of barbs. Posterior margin of tergite H protruded. Cupules *ia* on tergite D moderately elliptical; plates EF and H without cupules *im* and *ih*, respectively. Length of dorsal setae: *sc*2 17 (13–16), *c*1 21 (22–25), *c*2 20 (18–23), *d* 23 (24–26), *e* 17 (13–16), *f* 35 (35–40), *h*1 33 (31–34), *h*2 30 (30–32). Distances between dorsal setae: *sc*2–*sc*2 48 (47–50), *c*1–*c*1 49 (47–51), *c*2–*c*2 105 (103–115), *c*1–*c*2 28 (28–33), *d*–*d* 27 (27–34), *e*–*e* 71 (65– 70), *f*–*f* 56 (55–61), *e*–*f* 9 (7–9), *h*1–*h*1 28 (22–29), *h*2–*h*2 49 (47–50), *h*1–*h*2 11 (12–14).

*Idiosomal venter* (Figure 2b and Figure 4b). All ventral plates with numerous small dimples; posterior plate with higher density. Apodemes 1, 2 (ap1-2) and apsej well-developed and sclerotized, joined with appr; ap3 well-developed, ap4 moderately developed, both sclerotized, joined with appo; ap5 reduced. Posterior margin of posterior sternal plate weakly concave. All ventral setae pointed; setae 3*a*, 3*b* and 4*a* with a couple of barbs; other ventral setae smooth. Setae 1*b* pointed, with moderately large notch or slit in distal half (Figure 2c and Figure 4c); pseudanal setae *ps*1 thickened and characteristically curved, *ps*3 thin and pointed, *ps*2 absent. Posterior margin of aggenital plate hook-like. Anterior genital sclerite (ags) small, triangular, median genital sclerite (mgs) developed, asymmetric, posterior genital sclerite (pgs) well-developed, triangular, larger than other genital sclerites (Figure 2d and Figure 4d). Length of ventral setae: 1*a* 28 (26–29), 1*b* 26 (22–25), 2*a* 29 (23–27), 2*b* 28 (22–27), 3*a* 30 (26–29), 3*b* 30 (28–30), 3*c* 31 (24–26), 4*a* 35 (37–40), 4*b* 44 (41–45), 4*c* 33 (28–31), *ps*1 15 (13–16), *ps*3 20 (17–21).

*Legs* (Figure 3). *Leg I* (Figure 3a): setae formulae: Tr 1-Fe 3-Ge 4-TiTa 17(4) (number of solenidia in parentheses); tibiotarsus massive, terminating in large, sickle-like claw. All setae on tibiotarsus and genu smooth, pointed, except *u*′ on tibiotarsus, thickened, spiniform and blunt-ended; solenidion *ω*1 10 (8–9) thickened, digitiform, *ω*2 5 (5–6) small and uniform, *φ*1 6 (6–7) slender, *φ*2 6 (5–7), moderately clavate, slightly thicker than *ω*2. Setae *d* and *l*′ on femur thickened, seta *d* stout, hook-like, seta *l* rod-shaped; seta *v*′ on trochanter small, needle-like. *Leg II.* (Figure 3b): setae formulae: Tr 1-Fe 3-Ge 3-Ti 4(1)-Ta 6(1); with tongue-like empodium with pointed tip and a pair of slightly asymmetrical claws; all setae pointed, setae *pv*″ on tarsus, *v*′ and *v*″ on tibia with some barbs; other setae smooth; solenidion *ω* 14 (12–13) baculiform, *φ* 4 (4–5) small and weakly clavate. *Leg III* (Figure 3c): setae formulae: Tr 1-Fe 2-Ge 2-Ti 4(1)-Ta 5; with tongue-like empodium with pointed tip and pair of slightly asymmetrical claws; all setae pointed, except seta *pv*″ on tarsus, bearing 2–3 barbs; tibial setae same as leg II; solenidion *φ* 4 (4–5) small and weakly clavate. *Leg IV* (Figure 3d): setae formulae: Tr 1-Fe 2-Ge 1-Ti 4(1)-Ta 6; with slender and spike-like empodium and a pair of thinner claws than legs II-III; all setae smooth and pointed except seta *v*″ on tibia, with 1–2 barbs and blunt-ended; solenidion *φ* 4 (4–5) small and weakly clavate.

#### 3.1.1. Differential Diagnosis 

The new species is readily distinguishable from all other congeners (listed in Table 1) due to the presence of mgs. The remarkable shape of seta 1*b* (with a notch or slit in the distal half) has also never been described in any *Allopygmephorus*, but the same character was detected in *A. persicus* following a detailed microscopy (A. Khaustov; personal communication), although this was not depicted in its original description [36].

Outside the genus, the presence of mgs makes *A. coelostomus* sp. nov. somewhat close to *Protoallopygmephorus heteroceri* Khaustov and Sazhnev 2016. Nevertheless, the absence of setae *v*2 in *A. coelostomus* sp. nov. (present in *Protoallopygmephorus*), prominent solenidion *ω*1 (fused with tibiotarsus I in *Protoallopygmephorus*) and absence of setae *ps*3 (present in *Protoallopygmephorus*) in *A. coelostomus* sp. nov. makes it easily distinguishable from the genus *Protoallopygmephorus*.

#### 3.1.2. Etymology

The species epithet “*coelostomus*” refers to the generic name of the hydrophilid host, *Coelostoma* Brullé, 1835.


**Key to the world species of *Allopygmephorus***


1.With three solenidia on tibiotarsus I .......................................................................................................................................................................................... 2

-With four solenidia on tibiotarsus I ............................................................................................................................................................................................ 3

2.Seta *e* subequal to setae *h*1 and *h*2, pgs absence ........................................................................................................ *A. nanhuiensis* Gao, Zou and Ma, 1989

-Seta *e* about half the length of *h*1 and *h*2*,* pgs presence ............................................................................................................... *A. chinensis* Mahunka 1975

3.Seta 1a, modified, with very long barbs; Setae *f*, *h*1 and *h*2 blunt-ended .......................................................................................................................................................................................... *A. bakaninae* Khaustov and Ermilov, 2008

-Seta 1*a* normality; Setae *f*, *h*1 and *h*2 pointed or blunt-ended ...................................................................................................................................................................................... 4

4.Tibiotarsus I with a weak-stalked small claw ............................................................................................................................................................................... 5

-Tibiotarsus I with a normal or large claw .................................................................................................................................................................................... 6

5.Solenidion *ω*2 on tibiotarsus I long, approximately more than half the length of segment; *ω*2 > *ω*1, *φ*1 ≈ *φ*2 ........................ *A. cunae* Mahunka, 1970

-Solenidion *ω*2 on tibiotarsus I shorter than half the length of segment; *ω*2 ≈ *ω*1, *φ*1 > *φ*2 .......................................................... *A. brasiliensis* Mahunka, 1970

6.Setae *c*1, *c*2 and *d* basally thickened, bulbiform; *ps*2 absence .................................................................... *A. orientalis* Mahunka and Mahunka-Papp, 1988

-Setae *c*1, *c*2 and *d* not basally thickened, normality; *ps*2 absence or presence ......................................................................................................................... 7

7.Seta *c*1 considerably shorter than *c*2; 4*b* long, arising end of body; three pairs of *ps*; *h*1 and *h*2 blunt-ended, weakly barbed ............................................................................................. *A. tuberosus* Mahunka, 1969

-Seta *c*1 = *c*2 or *c*1 somewhat shorter than *c*2; 4*b* not arising end of body; three or two pairs of *ps*; *h*1 and *h*2 blunt-ended or pointed ...................................................................................................................................................................................................................................... 8

8.Seta 1*b* modified, with a notch or slit in the distal half; ags, mgs and pgs presence ................................................................. *A. coelostomus*
**sp. nov.**

-Seta 1*b* without big notch or slit, mgs absence, ags and pgs presence or absence .......................................................................................................................... 9

9.Seta 1*b* thickened and lanceolate; seta *h*2 blunt-ended, barbed ....................................................................................................................................... 10

-Seta 1*b* normality, barbed or smooth; seta *h*2 blunt-ended or pointed ............................................................................................................................... 11

10.Seta *h*1 blunt-ended; seta *v″* on femur II spiniform .......................................................................... *A. spinisetus* Khaustov and Sazhnev, 2016

-Seta *h*1 pointed; seta *v**″* on femur II unmodified ........................................................................................ *A. punctatus* Khaustov and Sazhnev, 2016

11.Seta *h*1 pointed; *h*1 and *h*2 subequal .......................................................................................... *A. persicus* Khaustov and Hajiqanbar, 2006

-Seta *h*1 blunt-ended, *h*1 and *h*2 subequal or *h*1 > *h*2 .......................................................................................................................................................... 12

12.Setae *h*1 and *h*2 subequal; solenidia *ω*2 > *φ*1 ≈ *φ*2 > *ω*1 on tibiotarsus I ........................................................ ... *A. baoshanensis* Gao, Zou and Ma, 1989

-Setae *h*1 > *h*2; solenidia otherwise ......................................................................................................................................................................................... 13

13.Solenidia *ω*1 > *φ*1 > *φ*2 > *ω*2 on tibiotarsus I; seta *e* shorter than half of the seta *f*........................................................................................................................................................................................................ *A. matthesi* Krczal, 1959

-Solenidia *ω*1 > *ω*2 ≈ *φ*2 > *φ*1 on tibiotarsus I; seta *e* about half the length of seta *f* .............................................................. *A. heterodactylus* Mahunka, 1973

### 3.2. Family Scutacaridae Oudemans, 1916

Genus *Archidispus* Karafiat, 1959.

Type Species: *Archidispus minor* (Karafiat, 1959), by original designation.

**Diagnosis**: Adult female. Seta 1*a* thickened, blunt-ended and smooth, setae 3*b*, 4*a* and 4*b* smooth, thickened at basal half and pointed, genital sclerites (at least ags and pgs) present, pharyngeal pumps II and III with short distance from each other, setae *ps*2 absent.

*Archidispus hydrophilus* Khadem-Safdarkhani and Hajiqanbar **sp. nov.** (Figure 5, Figure 6, Figure 7 and Figure 8).

Type Material.

**Holotype**: female (QMS 120025), ex. under sternite between legs coxae of *C. fabricii* (Montrouzier, 1860) (Coleoptera: Hydrophilidae); Loc. Vines Dr, Hawkesbury campus, Western Sydney University, Richmond, NSW, 33°36′45.6″ S 150°44′40.2″ E; Coll. A. Katlav; 12 February 2020.

**Paratypes**: Four females (QMS 120026-8; TMU SP202206-AK1), same data as holotype.

Description.

**Phoretic female**. Length of idiosoma 207 (223–229), width 182 (159–197).

*Gnathosoma* (Figure 5): length of gnathosoma 33 (32–34), width 28 (24–27). Gnathosomal capsule (Figure 5a) slightly longer than wide; with a weak longitudinal dorsal median apodeme in posterior half; dorsally with two pairs of subequal, pointed and moderately barbed cheliceral setae *ch*1 22 (20–22) and *ch*2 22 (19–21); *pp* evident in some specimens (not discernable in holotype); subcapitulum (Figure 5b) with one pair of pointed and smooth setae *su* 11, distance between seta *ch*1*–ch*1 13*, ch*2*–ch*2 22*, su–su* 10; palpi compressed to gnathosomal capsule, dorsally bearing two subequal, pointed, indistinctly-barbed setae *d*Fe 11 (10–12) and *d*Ge 12 (11–13), ventrally with a pair of prominent solenidia, palpi terminating in sclerotized claws and an accessory setigenous structure (*ass*) at base of claws. Pharyngeal system (Figure 5a) bearing one smooth and two striated pumps: pump 1 (php1) smallest and bow-like, pump 2 (php2) strongly developed, cylindrical, pump 3 (php3) developed and sub-pentagonal.

*Idiosomal dorsum* (Figure 6a and Figure 8a): Tergite C weakly dimpled, other tergites smooth; prodorsal shield (PrS) entirely covered by tergite C, with one pair of clavate trichobothria (*sc*1) (Figure 5c) bearing minute barbs, and two pairs of subequal setae *v*2 and *sc*2 (visible in pressure-mounted slides with removed tergite C); anterior of base of seta *sc*1 with two subequal, horn-like protrusions (Figure 5c); cupuli *ia* and *ih* visible on tergites C and H, respectively; setae *c*2 posterolaterally inserted into *c*1 and bearing a sclerotic alveolar canal; all dorsal setae moderately pointed and barbed; length of dorsal setae: *v*2 7 (7–8), *sc*2 6 (6–8), *c*1 36 (32–34), *c*2 39 (36–39), *d* 44 (40–42), *e* 49 (42–48), *f* 44 (36–38), *h*1 61 (61–62), *h*2 57 (53–56). Distances between dorsal setae: *c*1– *c*1 61 (59–63), *c*2–*c*2 117 (120–136), *c*1–*c*2 36 (36–38), *d*–*d* 92 (92–94), *e*–*e* 140 (138– 146), *f*–*f* 60 (58–62), *e*–*f* 41 (41–43), *h*1–*h*1 36 (34–38), *h*2–*h*2 101 (98–105), *h*1–*h*2 35 (30–34).

*Idiosomal venter* (Figure 6b and Figure 8b). Apodemes 1–4 (ap 1–4) and sejugal (apsej) well developed; ap1, ap2 and apsej fused with pre-sternal apodeme (appr); ap3 and ap4 fused with post-sternal apodeme (appo); secondary transverse apodeme (Sta) weakly developed, posteriad ap2, crossing appr; apodemes 5 (ap5) short, reduced to lateral fragments. Posterior margin of post-sternal plate flattened, weakly undulated; aggenital plate concave in posterior margin, bearing developed anterior genital sclerite (ags), median genital sclerite (mgs) and posterior genital sclerite (pgs). Coxisternal setae 1*b*, 2*a*, 2*b*, 3*a*, 3*c* and 4*c* setiform, slightly barbed and pointed. Setae 1*a* thickened, smooth and blunt-ended; setae 3*b*, 4*a* and 4*b* smooth, thickened at basal half and pointed; pseudanal setae *ps*1 and *ps*3 subequal, barbed and pointed, about twice the length of setae *ps*2; setae *ps*2 setiform, smooth and adjacent setae *ps*1. Length of ventral setae: 1*a* 10 (10–11), 1*b* 29 (29–32), 2*a* 25 (24–27), 2*b* 27 (29–31), 3*a* 28 (27), 3*b* 19 (20–21), 3*c* 31 (34–36), 4*a* 19 (19–22), 4*b* 33 (35–37), 4*c* 40 (39–43), *ps*1 33 (30–34), *ps*2 16 (12–15), *ps*3 31 (29–32).

*Legs* (Figure 7). *Leg I* (Figure 7a): distinctly thicker and shorter than other legs; setae formulae: Tr 1-Fe 3-Ge 4-TiTa 16(4) (number of solenidia in parentheses); tibiotarsus with a robust claw; all setae on tarsi smooth except setae *l*″ and *pv*″; *tc*″ and *ft*″ located apically and laterally, respectively, on a distinct pinnaculum; solenidia *ω*1 13 (12–15) and *ω*2 16 (15–18) baculiform, *φ*1 11 (10–11) finger-shaped, *φ*2 13 (11–14) very slender and baculiform; all setae on genu, femur and trochanter smooth; setae *d* on femur short and spined, medially serrate. *Leg II* (Figure 7b): setae formulae: Tr 1-Fe 3-Ge 3-Ti 4(1)-Ta 6(1); empodium developed; seta *u*′ smooth and indistinctly blunt-ended, *tc*″ smooth and pointed, setae *tc*′, *pv*″ and *pl*″ on tarsus slightly barbed; *ω* 14 (13–15) elongate and baculiform; setae *v*′ and *v*″ on tibia weakly barbed; *φ* 9 (7–9) weakly clavate; seta *l*″ on genu barbed; setae *d* on femur weakly blunt-ended; seta *v*′ on trochanter moderately barbed and indistinctly blunt-ended. *Leg III* (Figure 7c): setae formulae: Tr 1-Fe 2-Ge 2-Ti 4(1)-Ta 6; empodium developed; *u*′ shortest, smooth and indistinctly blunt; *tc*″ smooth and pointed; *pv*″ smooth and indistinctly blunt-ended; other tarsal setae moderately barbed; on tibia setae *v*′ and *v*″ barbed and indistinctly blunt-ended, *d* and *l*′ smooth; solenidion *φ* 6 (7) uniform; on genu *v*′ and *l*′ smooth; on femur seta *d* barbed and pointed, *l*′ smooth and slightly blunt-ended; on trochanter seta *v*′ barbed and pointed. *Leg IV* (Figure 7d): the longest leg; setae formulae: Tr 1-Fe 2-Ge 1-Ti 3(1)-Ta 6; proximal portion of tarsus elongated, tapering to apex; all setae blunt-ended, *u*′ smooth *tc*″ smooth, seta *pl*″ smooth and rod-like; other tarsal setae barbed; on tibia all setae barbed and indistinctly blunt-ended, solenidion *φ* 6 (6–7) thin and uniform; on genu seta *v*′ barbed and pointed; on femur setae *d* and *v*′ barbed and weakly blunt-ended; on trochanter seta *v*′ sparsely barbed and pointed.

#### 3.2.1. Differential Diagnosis

This species is readily distinguishable from all other species of the genus due to the shape of seta 1*a* (thickened, blunt-ended and smooth) and presence of mgs; regardless, *A. hydrophilus* sp. nov. is most similar to *A. kazuyoshikurosai* Khaustov based on setae 3*b*, 4*a* and 4*b* being thickened in the basal half and thin in the distal half; however, it differs from the new species as all dorsal setae are moderately pointed (most dorsal setae are distinctly blunt-ended in *A. kazuyoshikurosai*), and setae 1*a* being short, spiniform and blunt-ended (setae 1*a* longer and beak-like in *A. kazuyoshikurosai*), setae *pl*″ on tarsus IV with blunt-tipped rod (pointed in *A. kazuyoshikurosai*).

#### 3.2.2. Etymology

The species epithet ‘*hydrophilus’* means water-loving and refers to the humid environment in which the hydrophilid host beetle of this species and most likely the species itself dwell.

## 4. Discussion

The host beetle genus *Coelostoma* shows the greatest diversity in the Afrotropical and Oriental regions, with a few species in the Palearctic and Australian realms [49,50,51,52]. Although many hydrophilid beetles are aquatic, some are semiaquatic, including *Coleostoma*, which resides in wet grasslands near riverbanks and ponds. Here, they live under wet rocks or among grass roots, where they feed on decomposing organic matter [51], which is also an ideal substrate for the growth of fungal mycelia—the likely food source for the phoretic *Allopygmephorus* and *Archidispus* mites. Many *Coelostoma* species are nocturnal: during the day, they usually hide under moss or the roots of plants growing next to the watercourse, and may be found feeding on wet and submerged surfaces, including those of wet rocks and artificial concrete surfaces, at night. Some species may be collected from mud or from underneath wet leaf litter; few species are only found in interstitial habitats under stones and among gravel at the sides of stony riverbeds [51]

*Allopygmephorus* is well-known from sodden and rotting vegetation matter or from the beetles that frequent these habitats, especially Hydrophilidae and Heteroceridae (Table 1). The genus is widespread, occurring in the Afrotropical, Indo-Malayan, Neotropical and Palearctic regions. Its discovery in the Australian realm is, thus, not surprising, although its presence on *C. fabricii,* the sole Australian representative of this large Afrotropical–Oriental beetle genus, suggests it may have arrived, together with its carrier, from the Oriental realm. However, this hypothesis makes the unconvincing assumption that *A. coelostomus* has a close host relationship with its carrier. Previous collections of *Allopygmephorus* suggest a low host specificity in the group, with mites riding on several families of beetles that presumably visit the same habitat (Table 1).

Scutacarid mites generally use their insect hosts for phoretic dispersal between ephemeral habitats; however, some can be inquilines of their hosts’ nests. It is hypothesized that some species may play a sanitary role in their hosts’ nests [24]. Some genera show strong preferences towards special insect groups, while others are comparatively less host-specific [24]. *Archidispus* is a group of scutacarid mites with a strong preference for phoresy on carabid beetles [24]. Records from other beetle species are rare but do include a record of *A. bembidii* from *Coelostoma* hydrophilid beetles [9]. Similarly to *Allopygmephorus*, numerous species of *Archidispus* utilize several carrier species and it remains unknown if *A. hydrophilus* is specific to *C. fabricii* or is opportunistically phoretic on various semiaquatic beetles. Broader surveys are needed to provide more information and uncover more of Australia’s vast undescribed fauna of phoretic and other mites.

## Figures and Tables

**Figure 1 insects-13-00483-f001:**
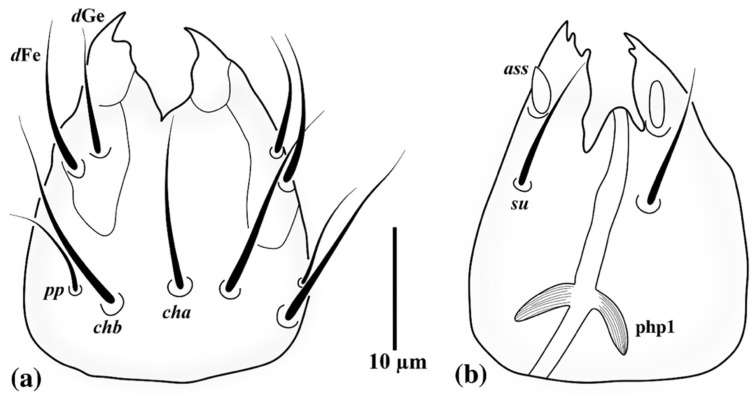
*Allopygmephorus coelostomus* sp. nov. (**a**) dorsal view of gnathosoma; (**b**) ventral view of gnathosoma.

**Figure 2 insects-13-00483-f002:**
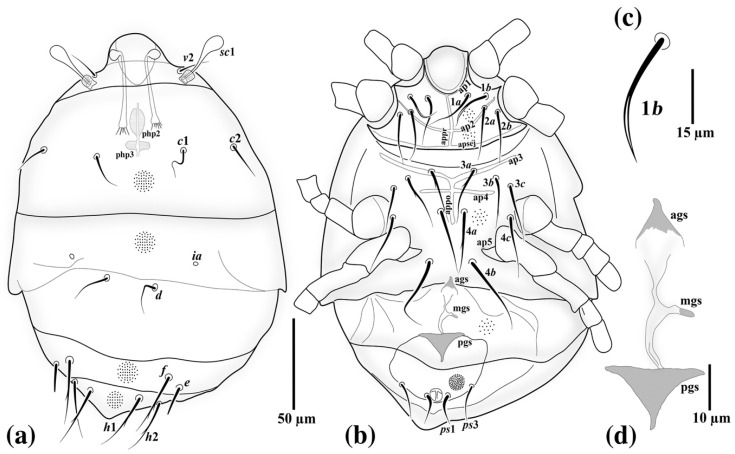
*Allopygmephorus coelostomus* sp. nov. (**a**) dorsal view of body; (**b**) ventral view of body; (**c**) seta 1*b*; (**d**) genital sclerites.

**Figure 3 insects-13-00483-f003:**
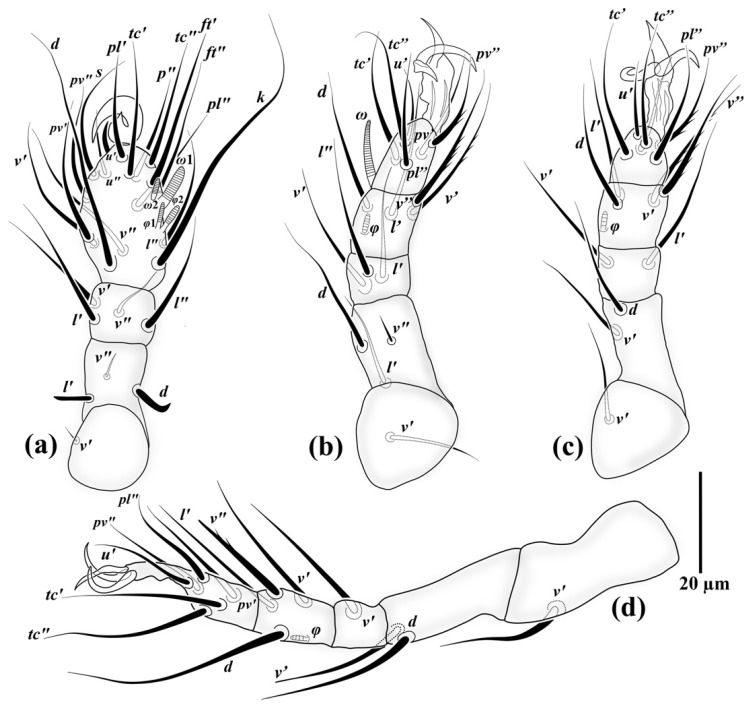
Right legs of *Allopygmephorus coelostomus* sp. nov. (**a**) leg I; (**b**) leg II; (**c**) leg III; (**d**) leg IV.

**Figure 4 insects-13-00483-f004:**
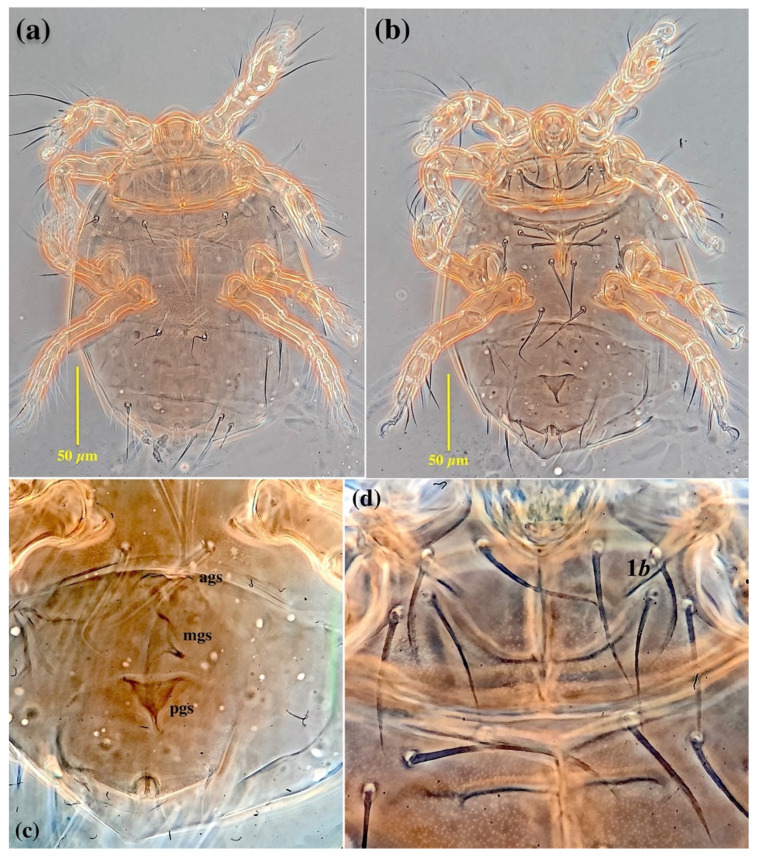
Micrograph of *Allopygmephorus coelostomus* sp. nov., phoretic female (**a**) general view dorsally; (**b**) general view ventrally; (**c**) genital sclerites; (**d**) setae 1*b* with a notch or slit in the distal half.

**Figure 5 insects-13-00483-f005:**
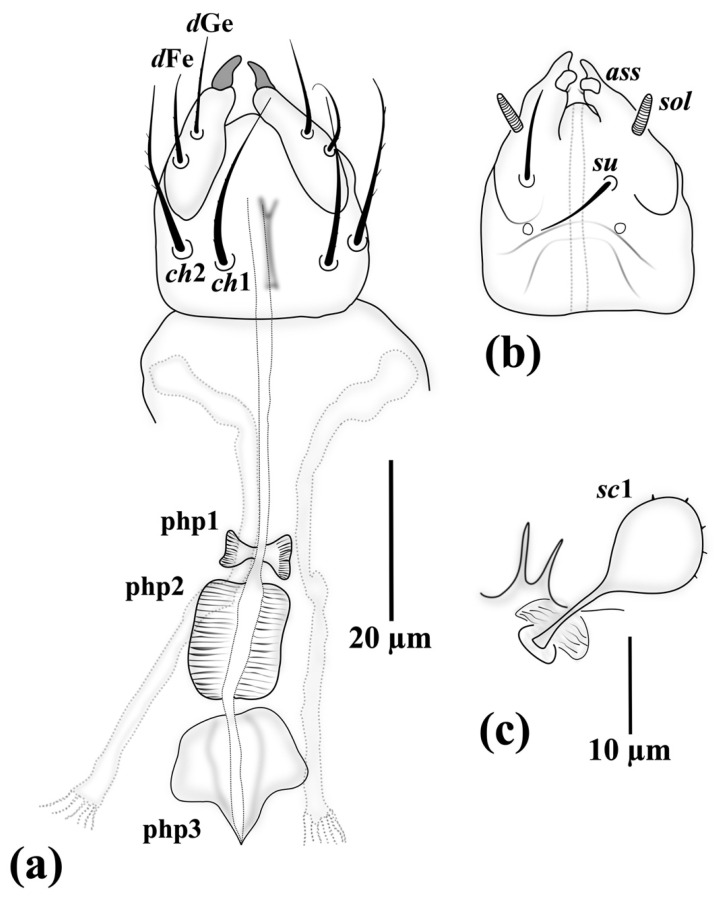
*Archidispus hydrophilus* sp. nov. (**a**) dorsal view of gnathosoma; (**b**) ventral view of gnathosoma; (**c**) seta *sc*1.

**Figure 6 insects-13-00483-f006:**
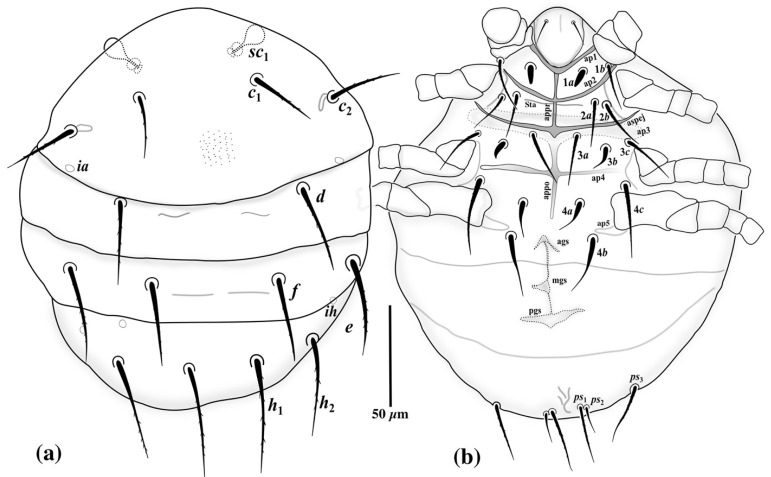
*Archidispus hydrophilus* sp. nov. (**a**) dorsal view of body; (**b**) ventral view of body.

**Figure 7 insects-13-00483-f007:**
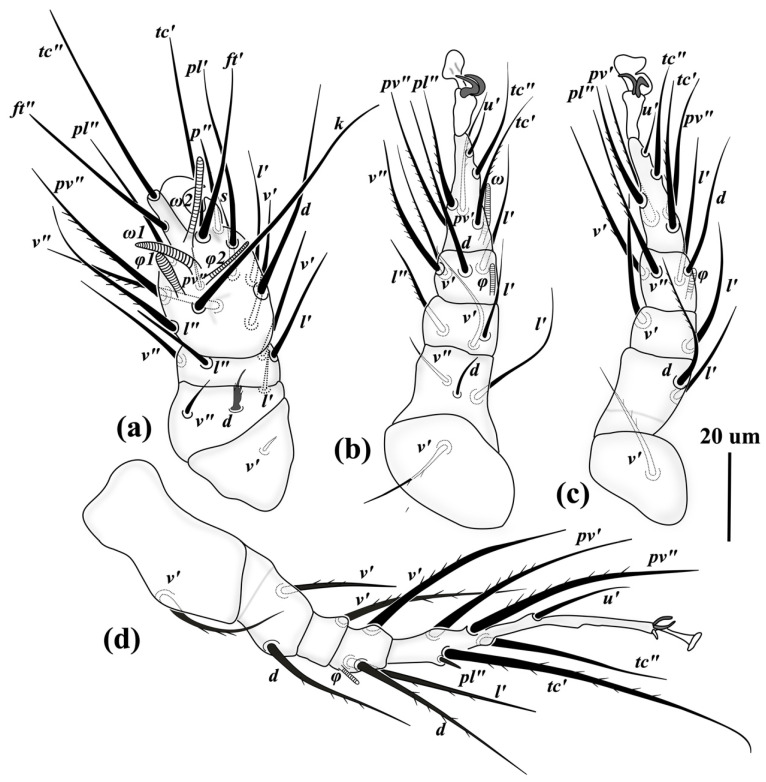
Right legs of *Archidispus hydrophilus* sp. nov. (**a**) leg I; (**b**) leg II; (**c**) leg III; (**d**) leg IV.

**Figure 8 insects-13-00483-f008:**
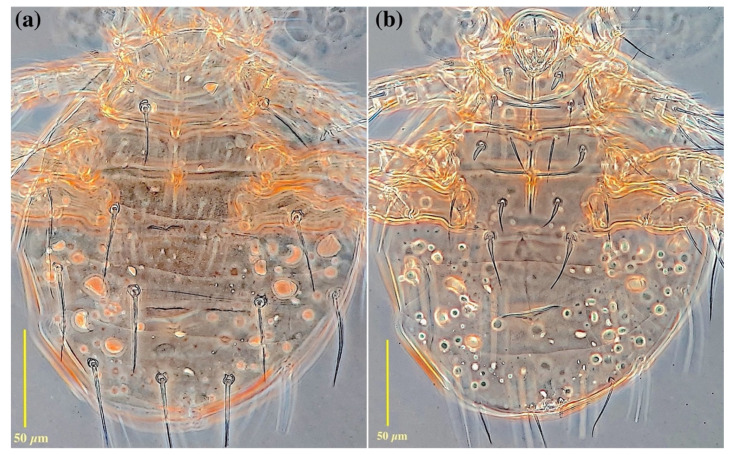
Micrograph of *Archidispus hydrophilus* sp. nov., phoretic female (**a**) dorsal general view; (**b**) ventral general view.

**Table 1 insects-13-00483-t001:** Information on the distribution and host and/or habitat associations of all known species of *Allopygmephorus*.

Mite Species	Distribution	Host and/or Habitat	Reference
*Allopygmephorus bakaninae* Khaustov and Ermilov, 2008	Russia	In wet soil	[37]
*Allopygmephorus baoshanensis* Gao, Zou and Ma, 1989	China	Unknown	[38]
*Allopygmephorus brasiliensis* Mahunka, 1970	Brazil	Litter and humus among roots surrounded by water	[39]
*Allopygmephorus chinensis* Mahunka 1975	Hong Kong	Wet and rotting leaves	[40]
*Allopygmephorus cunae* Mahunka, 1970	Brazil	Very wet decaying grass and detritus on river shore	[39]
*Allopygmephorus heterodactylus* Mahunka, 1973	Ghana	Unidentified beetle; *Cercyon* sp. (Col.: Hydrophilidae)	[41]
*Allopygmephorus matthesi* Krczal, 1959	Germany; Tanzania; Iran	*Berosus luridus* (Linnaeus, 1760), *Coelostoma orbiculare* (Fabricius, 1775), *Enochrus quadripunctatus* (Herbst, 1797), *Helochares griseus* (Fabricius, 1787), *H. lividus* (Forster, 1771), *Hydrobius fuscipes* (Linnaeus, 1758), *Hydrophilus caraboides* (Linnaeus, 1758), *Philydrus melanocephalus* Kuwert, 1888 (Col.: Hydrophilidae); *Heterocerus marginatus* (Fabricius, 1787) (Col.: Heteroceridae); *Dryops auriculatus* (Geoffroy, 1785) (Col.: Dryopidae); *Xyleborus* sp. (Col.: Curculionidae: Scolytinae)	[41,42,43]
*Allopygmephorus nanhuiensis* Gao, Zou and Ma, 1989	China	Unknown	[38]
*Allopygmephorus orientalis* Mahunka and Mahunka-Papp, 1988	Malaysia	Rotting wood and leaves	[44]
*Allopygmephorus persicus* Khaustov and Hajiqanbar, 2006	Iran	*Cercyon laminates* Sharp, 1873, *Enochrus bicolor* (Fabricius, 1792) (Col.: Hydrophilidae)	[36,45]
*Allopygmephorus punctatus* Khaustov and Sazhnev, 2016	Russia; Iran	*Heterocerus fenestratus* (Thunberg, 1784), *Heterocerus flexuosus* Stephens, 1828 (Col.: Heteroceridae); *Drasterius bimaculatus* (Rossi, 1790) (Col.: Elateridae); *Augyles* sp. (Col.: Heteroceridae)	[46,47]
*Allopygmephorus spinisetus* Khaustov and Sazhnev, 2016	Russia Iran	*Heterocerus fenestratus* (Col.: Heteroceridae); *Drasterius bimaculatus* (Col.: Elateridae); *Augyles* sp. (Col.: Heteroceridae)	[46,47]
*Allopygmephorus tuberosus* Mahunka, 1969	Bolivia	Unknown	[48]
*Allopygmephorus coelostomus* sp. nov.	Australia	*Coelostoma* (*Coelostoma*) *fabricii* (Col.: Hydrophilidae)	This study

## Data Availability

This published work and the nomenclatural acts it contains have been registered in ZooBank, the online registration system for the ICZN (International Code of Zoological Nomenclature).

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
