# Peer review of "Two New Phoretic Species of Heterostigmatic Mites (Acari: Prostigmata: Neopygmephoridae and Scutacaridae) on Australian Hydrophilid Beetles (Coleoptera: Hydrophilidae)†"

_insects, 2022, doi:10.3390/insects13050483_

Round 1

Reviewer 1 Report

In the reviewed MS an international group of scientists including acarologists from Iran and Australia describe 2 new heterostigmatan species from Australia. The MS is interesting to read. The introduction is informative and enriched with large number of adequate references. The drawings are of high quality. There are some typos in the text (I colored them yellow in the attached pdf file). Some setal abbreviations should be checked in order to make all of them uniform (italic) both in the text and in the figures. May be the authors could insert several high-quality microphotographs of the mites and the beetles, such graphical data would make the MS more attractive. I found Discussion to be written stylistically a bit like a "Remark" which we often see in zoological papers e.g. in Zootaxa. May be the authors could slightly reformat the Discussion. In my opinion this MS meets all required criteria and I consider it to be appropriate for publishing in Insects.

Author Response

Reviewer 1: There are some typos in the text (I coloured them yellow in the attached pdf file).

Authors: These all have been corrected in the respective sections raised by the reviewer.

Reviewer 1: Some setal abbreviations should be checked in order to make all of them uniform (italic) both in the text and in the figures.

Authors: Setal abbreviations have now been double-checked and made uniform throughout the descriptions.

Reviewer 1: May be the authors could insert several high-quality microphotographs of the mites and the beetles, such graphical data would make the MS more attractive.

Authors: We agree, these have now been added – please see figures 4 and 8.

Reviewer 1: I found Discussion to be written stylistically a bit like a "Remark" which we often see in zoological papers e.g. in Zootaxa. May be the authors could slightly reformat the Discussion.

Authors: We thank the reviewer’s suggestion regarding the ‘Discussion’ structure, and we have made some readjustments in this section to comply with this suggestion. Please also note this section mainly deals with some important biogeographical and host range information of the host beetle and associated mite species, and we believe they need to be placed in a separate section like a “Discussion” rather than a short taxonomic remark that are commonly provided at the end of descriptions in other systematic papers.

Reviewer 2 Report

This manuscript described two new species in Australia and should be published. However, I am concerned about over-estimation regarding a role of the mite species. Because authors did not conduct experiments/ observation to clarify the role of two mite species, some states must be deleted and/or weakened. 

L26-27: This sentence is an oversimplification and should be deleted, as the study did not conduct any indirect or experimental studies on the potential role of the mites, nor did it reveal the ecology of them.

L56-58: Baumann (2018) did not suggest the sanitary role of the mite group based on observational studies of “these mites” but different mite groups such as mesodtigmatids. Even, the idea that a mite species associated in a nest remove harmful fungi was not proposed with scientific observations but only a guess.

L73-74: The correct sentence may be “their phoretic associations are mostly unknown” because phoretic associations between Scutacarus hydrophilus and phorid or sciarid flies have been known.

L183: Is this about Allopygmephorus or Archidispus?

L194-: I do not think the table is necessary.

L330: “just” means “only one single species”? Please make it clearer.

L334: I do not agree. Subaquatic habitats are suitable for bacteria rather than fungi.

Author Response

Reviewer 2: L26-27: This sentence is an oversimplification and should be deleted, as the study did not conduct any indirect or experimental studies on the potential role of the mites, nor did it reveal the ecology of them.

Authors: We have deleted this section as suggested by the reviewer. 

Reviewer 2: L56-58: Baumann (2018) did not suggest the sanitary role of the mite group based on observational studies of “these mites” but different mite groups such as mesodtigmatids. Even, the idea that a mite species associated in a nest remove harmful fungi was not proposed with scientific observations but only a guess.

Authors: In agreement with the comment of the reviewer we have omitted this section.

Reviewer 2: L73-74: The correct sentence may be “their phoretic associations are mostly unknown” because phoretic associations between Scutacarus hydrophilus and phorid or sciarid flies have been known.

Authors: This has been adjusted as suggested by the reviewer

Reviewer 2: L183: Is this about Allopygmephorus or Archidispus?

Authors: This section is about “Allopygmephorus”. This error has now been adjusted.

Reviewer 2: L194-: I do not think the table is necessary.

Authors: Although we appreciate the reviewer’s viewpoint, we believe this table provides for the first time an overview of the distribution and host and/or habitat associations of all known species of Allopygmephorus. We believe useful to keep this because otherwise readers would have to go through the arduous task of retrieving these data from many different resources. Thus, we would like to keep this table in this paper.

Reviewer 2: L330: “just” means “only one single species”? Please make it clearer.

Authors: Yes, this means “only one single species” and we have now clarified this.

Reviewer 2: L334: I do not agree. Subaquatic habitats are suitable for bacteria rather than fungi.

Authors: I think there is a misunderstanding here – our manuscript refers to the host beetle species to be semi-aquatic, and the fungi develop in the moist/semi-aquatic environment. We do not use the term sub-aquatic in the manuscript. However, there are also references in the literature that showing that fungal biomass is highly reliant on high humidity or moisture content, and they can grow well under aquatic and subaquatic semi-aquatic conditions. Please see Hans-Peter Grossart et al. (2019) in Nature Reviews Microbiology.

Reviewer 3 Report

The article is well conceptualized and nicely written.  The quality of the illustrations is beautiful where a minor change is suggested. I would like to propose including some of the DIC images in support of the illustration and to give a test of type materials. The overall scientific merit is high and certainly enriches the knowledge of the science of acarology.

Author Response

REVIEWER 3

Reviewer 3: I would like to propose including some of the DIC images in support of the illustration and to give a test of type materials.

Authors: We agree, these have now been added – please see figures 4 and 8.

We have also incorporated all adjustments/corrections in the manuscript version annotated by Reviewer in  the revised version of our ms.

Reviewer 4 Report

This is potentailly good paper, well thought out and presenting interesting discoveries, but plagued by a host of poor editing issues. Details are in attached MS supplied by remarks. The artwork is very fine, though photographs would make it even better.

Potential problem with too many authors of species' names should be addressed.

Author Response

REVIEWER 4

Reviewer 4: Details are in attached MS supplied by remarks.

Reviewer 4: We have made all adjustments/corrections in the manuscript in the sections annotated/highlighted by the reviewer.

The artwork is very fine, though photographs would make it even better.

Authors: We agree, these have now been added – please see figures 4 and 8.